# Methionine Restriction and Cancer Biology

**DOI:** 10.3390/nu12030684

**Published:** 2020-03-03

**Authors:** Desiree Wanders, Katherine Hobson, Xiangming Ji

**Affiliations:** Department of Nutrition, Byrdine F. Lewis College of Nursing and Health Professions, Georgia State University, Atlanta, GA 30302, USA

**Keywords:** methionine, cancer, methylation, amino acids, glutathione

## Abstract

The essential amino acid, methionine, is important for cancer cell growth and metabolism. A growing body of evidence indicates that methionine restriction inhibits cancer cell growth and may enhance the efficacy of chemotherapeutic agents. This review summarizes the efficacy and mechanism of action of methionine restriction on hallmarks of cancer in vitro and in vivo. The review highlights the role of glutathione formation, polyamine synthesis, and methyl group donation as mediators of the effects of methionine restriction on cancer biology. The translational potential of the use of methionine restriction as a personalized nutritional approach for the treatment of patients with cancer is also discussed.

## 1. Introduction

Cancer is the second leading cause of death in the United States, with an estimated 1.7 million new cases in the U.S. in 2019 alone [1]. Globally, the amount of cancer-related deaths outnumbered any other disease-related death, rising higher than coronary heart disease, or stroke [2]. Our understanding of cancer’s unique programming has led to the definition of the six hallmarks of cancer. These characteristics include cancer cells’ ability to (1) maintain proliferative signaling, (2) bypass growth suppressors, (3) resist apoptosis, (4) enable replicative immortality, (5) induce angiogenesis, and (6) initiate invasion and metastasis [3]. Once cancer has metastasized, the risk of mortality increases, making metastatic cancer particularly difficult to treat [4]. The current treatment options for cancer vary based on the type of cancer, as well as progression, and includes chemotherapy, radiation therapy, immunotherapy, and targeted therapy. Traditional chemotherapy has been the primary treatment for cancer based on its ability to inhibit rapidly dividing cancer cells [5]. While this treatment has increased the lifespan of many patients, the toxic effects on normal cells have detrimental health consequences. According to a recent national survey, 48% of people undergoing chemotherapy reported side effects including pain, nausea, and vomiting, while fatigue was reported in 80% of patients [6]. Additionally, the efficacy of chemotherapy can be limited due to drug resistance that may occur in some patients after several cycles of treatments [7]. Recently there has been much interest in shifting cancer treatment from cytotoxic non-specific agents to highly selective, mechanism-based therapeutics, thus paving the way for advanced nutritional therapies [8]. 

The dependence of many tumor cells on an exogenous source of the sulfur amino acid, methionine, [9,10,11] makes dietary methionine restriction (MR) an exciting potential tool in the treatment of cancer. Proliferation and growth of several types of cancer cells are inhibited by MR, while normal cells are unaffected by limiting methionine as long as homocysteine is present [9]. In addition to inhibiting cancer cell growth, MR has been shown to enhance efficacy of chemotherapy and radiation therapy in animal models [12]. Herein, we provide a review of the literature regarding efficacy and mechanism of action of methionine restriction on cancer cell growth in vitro and in vivo. We also provide insight into the translational potential of methionine restriction as a cancer treatment in humans. 

## 2. Methionine Restriction

Methionine is an essential amino acid, and as such it must be consumed in the diet to sustain life. Despite consumption of methionine being essential for survival, studies have shown that limiting methionine in the diet of animals or in cell culture media provides metabolic benefits such as decreasing adiposity [13,14,15,16,17,18,19,20,21], increasing insulin sensitivity [15,16,17,20,21,22], decreasing inflammation [23,24,25], and oxidative stress [26,27,28,29,30,31,32,33], and extending lifespan [34,35,36]. In fact, rats fed a diet with 80% less methionine lived 40% longer than rats fed a control diet [34]. Subsequent studies found that dietary MR was also effective at extending lifespan in outbred mice [35] and several rat strains with different age-related pathologies [36]. The lifespan-extending effects of MR have been attributed to a number of different mechanisms, including MR-induced reductions in oxidative stress [30] and inflammation [23], alterations in autophagy [37], and increases in cardioprotective hormones [15,38,39]. Another mechanism by which MR may extend lifespan is by providing a reduction in cancer incidence and overall reduction in cancer mortality.

The ability of MR to improve insulin sensitivity and reduce adiposity may be directly related to its anti-cancer potential as there are several types of cancer that are closely linked to obesity and insulin resistance [40,41,42,43] and the anti-cancer effects of MR may be secondary to its ability to reduce adiposity and increase insulin sensitivity. In mice, eight weeks of dietary MR produced a 3.1-fold increase in whole-body insulin sensitivity and an increase in tissue-specific glucose uptake measured during a hyperinsulinemic-euglycemic clamp [17,22]. Additionally, MR enhanced insulin-stimulated Akt phosphorylation in liver, muscle, and brown and white adipocytes in mice [22]. At least part of the insulin-sensitizing effect of MR can be attributed to its ability to reduce body weight and adiposity. However, limiting media methionine concentrations also enhances insulin signaling in HepG2 cells, indicating cell-autonomous effects of MR [22]. Of note, methionine restriction is effective when the non-essential amino acid, cysteine, is absent from the diet or media. Inclusion of cysteine reverses the effects of MR on metabolism and antioxidant status [16,44,45]. 

## 3. Methionine Metabolism

While methionine is involved in many biological processes, this review will highlight three major functions of methionine with relevance to cancer biology: (1) glutathione formation, (2) polyamine synthesis, and (3) methyl group donation (Figure 1).

### 3.1. Glutathione Formation 

Glutathione (GSH) is a thiol antioxidant that scavenges reactive oxygen species (ROS), resulting in the formation of oxidized glutathione (GSSG) [46]. Decreased amounts of GSH and a decreased GSH/GSSG ratio in tissues are biomarkers of oxidative stress [46]. Intracellular ROS can activate the PI3K pathway, which has been linked to increased cancer cell growth [47]. Chronic oxidative stress may lead to chronic inflammation and cancer development and progression [48].

The pathways involved in GSH production are regulated by many factors, including methionine. In the trans-sulfuration pathway, methionine is the precursor for cysteine which is essential for the formation of GSH [49]. Surprisingly, when methionine was restricted by 80% in the diet of rats, the level of GSH in the blood actually increased due to adaptations in sulfur-amino acid metabolism [31,50]. However, GSH concentrations were reduced in the liver [31,50]. Despite the MR-induced reduction in hepatic GSH, MR does not increase oxidative stress, in part because MR enhances antioxidant capacity and increases proton leak in the liver, likely decreasing ROS production [51]. The ability of MR to increase GSH levels in red blood cells is significant because of the role of GSH in the neutralization of ROS [52]. Several animal studies have shown restricting dietary methionine by just 40% reduces mitochondrial ROS production in several tissues, resulting in reduced mitochondrial DNA oxidative damage in vivo [30,53,54].

### 3.2. Polyamine Synthesis

Polyamines are small, naturally occurring cations that are essential in preserving chromatin structure, regulating ion-channels, and maintaining membrane stability [55]. Methionine is a precursor for the polyamines, spermidine and spermine. S-adenosylmethionine decarboxylase (SAMDC) is the key enzyme involved in polyamine biosynthesis. The product of its catalytic reaction, decarboxylated S-adenosylmethionine (dcSAM), serves as an aminopropyl donor in the biosynthesis of spermidine and spermine [56]. While spermidine is synthesized in most cells, spermine is specifically formed from the decarboxylation of S-adenosyl methionine by SAMDC in eukaryotic cells. Polyamines play a role in protein synthesis by regulating transcriptional and translational stages [57]. Polyamines are involved in the growth and proliferation of eukaryotic cells. Therefore, during polyamine depletion, there is a disruption in the cell cycle, inducing apoptosis [58,59]. Studies have shown that elevated levels of polyamines are associated with increased tumor growth [57]. Inhibitors of polyamine metabolism, such as alpha-difluoromethylornithine, lead to a reduction in polyamine production, and a disruption of the cell cycle and DNA synthesis in cancer cells [60,61]. In addition, other studies suggest that the mechanism of growth inhibition includes a downregulation of polyamine biosynthesis and the induction of the cyclin-dependent kinase inhibitor, p21 [59,62]. In cancer, there is often a disruption in polyamine metabolism leading to an increase in oxidative damage [56]. Polyamine metabolism is a potential target for treatment of several types of cancers [57]. Given that polyamine synthesis is dependent upon methionine, MR may be a novel approach to inhibit cancer cell growth by downregulating polyamine formation. 

### 3.3. DNA Methylation

DNA methylation is one of the most well characterized epigenetic modifications. Methylation takes place on most CpG dinucleotides occurring on 70% of cytosine bases. Cancer has been associated with both global DNA hypomethylation and gene-specific hypermethylation [63]. Hypermethylation of CpG islands in gene promoter regions may cause aberrant silencing of transcription and is a mechanism for downregulation of tumor-suppressor genes [64]. In fact, CpG island hypermethylation of specific genes is a hallmark in many cancer types [63,65]. S-adenosyl methionine (SAM) is the universal methyl donor for the methylation of DNA, RNA, histones, phospholipids, catecholamines, and proteins [66]. The methylation of various biomarkers (TFPI2, SEPT9, GSTP1, MGMT) in the presence of SAM have been shown to cause changes in tumor growth or suppression [67]. Methylation is a reversible modification, suggesting that dietary MR has the potential to affect cancer development and progression through modifying DNA methylation patterns. 

In general, aging is associated with global DNA hypomethylation despite some regions of specific genes becoming hypermethylated. It has been proposed that methionine metabolism is sensed by DNA methyltransferases that could alter methylation and lifespan [68]. Interestingly, in rodents, dietary methionine restriction has varying effects on global DNA methylation depending on the age of the animal. Twelve weeks of dietary MR increased global DNA methylation in the liver of old (age 1 year at start of intervention) mice, but had no effect on hepatic global DNA methylation in young (age 6 weeks at start of intervention) mice [69]. These data suggest that altered DNA methylation may be an important factor in the health benefits of MR [69].

## 4. Methionine Restriction and Cancer

### 4.1. Overview

In 1959, one of the early studies conducted in methionine restriction evaluated several outcomes produced from diets lacking specific amino acids. The study was conducted on rats fed isocaloric diets that were complete in all amino acids or devoid of one essential amino acid [70]. After transplantation of the Walker tumor, and 10-day preparative diet, rats were divided into different groups. Each group was fed a specific diet with different amino acid compositions for 5 days. While the initial aim of this study was to distinguish between two opposing views on nitrogen balance and amino acid restriction, the results showed a significant reduction in tumor growth in the rats fed diets lacking either methionine, valine, or isoleucine [70]. 

A subsequent study published in 1974 focused on methionine specifically [71]. This study was conducted on tissue cultures including W-256 (a rat breast cancer cell line), L1210 (a mouse lymphatic leukemia cell line), J111 (a human leukemia cell line), liver epithelial and liver fibroblasts of rats, skin fibroblasts of mice, and human breast and prostate cells that were normal or malignant [71]. The cells were cultured in folic acid- and cyanocobalamin-rich medium that either contained methionine or was methionine-free with a homocysteine supplement. Despite the media containing other methyl donors, the growth of the malignant cells was significantly impaired in the methionine-depleted media, while the normal cell growth was unchanged. These effects were attributed to the ability of normal cells to recycle homocysteine through methionine synthase to supply methionine endogenously. While this is true for normal cells, malignant cells lack the enzyme required to recycle homocysteine therefore giving methionine restriction the capacity to alter cancer cells while maintaining normal, healthy cells [72,73]. This enables the possibility that methionine restriction, as a therapeutic, may be able to specifically target cancer cells, preventing off-target effects on normal cellular processes. The following sections of this review provide an overview of the literature regarding methionine restriction and specific cancer types, including prostate, breast, and colorectal cancers (Table 1).

### 4.2. Methionine Restriction and Prostate Cancer 

Prostate cancer is the second leading cause of cancer death among adult men in the US and current treatment options include hormonal therapy to reduce testosterone levels, radiation therapy, or surgical procedures [90]. While there are treatment options available for prostate cancer, there are no known interventions to prevent the development of prostate cancer. Using a well-characterized mouse model for prostate cancer (Transgenic Adenocarcinoma of the Mouse Prostate; TRAMP), it was shown that dietary MR inhibits prostate cancer development especially in the anterior and dorsal lobes of the prostate, where the most severe lesions are found [82]. While the mechanism by which MR inhibits prostate cancer development is not known, evidence suggests that MR may work by inhibiting prostate cancer cell proliferation, inhibiting the insulin/IGF-1 axis, or by reducing polyamine synthesis [82]. The cells of the prostate produce high levels of polyamines and inhibition of polyamine synthesis is effective at suppressing tumor growth in prostate cancer [91]. Given the dependence of polyamine synthesis on methionine, the polyamine biosynthetic pathway may be a primary target of MR in prevention and/or treatment of prostate cancer.

Another target of MR in prostate cancer cells is thymidylate synthase (TS). Thymidylate synthase is the enzyme that catalyzes the methylation of deoxyuridylic acid during nucleotide biosynthesis and is thus an important target for cancer treatment. The chemotherapy drug, 5-fluorouracil (5-fu), inhibits TS activity by disrupting action of TS, causing DNA and RNA damage, making 5-fu an effective and commonly used cancer treatment [74]. However, 5-fu has also been reported to increase TS protein expression, resulting in 5-fu drug resistance [92]. Interestingly, several studies have shown that MR and 5-fu have synergistic anti-cancer effects [12,83,84,87]. MR selectively reduces TS activity in prostate cancer cells by ~80% within 48 h, but does not affect TS activity in normal prostate epithelial cells [74]. Importantly, MR also reduces TS protein expression, potentially explaining the synergy between MR and 5-fu [74]. That MR also reduces TS protein expression may make MR an attractive treatment alongside 5-fu to help combat resistance to 5-fluoruracil.

Methionine restriction has been shown to induce apoptosis in the human prostate cancer cell lines, PC3 and DU145 [75,76,77]. MR inhibits Raf and Akt oncogenic pathways, while increasing caspase-9 and the mitochondrial pro-apoptotic protein, Bak [75,76]. Restricting media methionine concentrations damages mitochondrial integrity, leading to apoptosis in both prostate cancer cell lines [75,76]. Additionally, energy production was impaired and ROS production was decreased. Caspase-dependent and -independent apoptosis was observed in response to MR [75,76]. Other studies have identified that c-Jun N-terminal kinases (JNK1) is a critical regulator of MR-induced apoptosis in prostate cancer cells [78].

In another study, PC-3, DU-145, and LNCaP human prostate cancer cell lines were cultured in complete- or methionine-free media and methionine dependency was evaluated [77]. The results showed that PC-3 is completely methionine-dependent, while DU-145 cells were mildly dependent, and LNCaP cells were almost completely methionine-independent. These data indicate that the responses to methionine restriction vary across different cancers, although MR inhibited growth of all three cancer cell lines [77]. The mechanisms by which MR reduced cancer growth also differed between the cell lines, with MR upregulating p21 and p27 (cell cycle inhibitors that halt cell cycle progression) in LNCaP cells, but only increasing p27 in PC-3 cells [77]. Further, the PC-3 cells began to undergo apoptosis within six days of MR, whereas the LNCaP cells were relatively resistant to MR-induced apoptosis [77]. Together, these data indicate a precision diet such as MR may benefit a subpopulation of patients with prostate cancer.

### 4.3. Methionine Restriction and Breast Cancer

Breast cancer is the second most common form of cancer diagnosed in women. Depending on the type of breast cancer, treatment options include surgery to remove the cancer or the entire breast, chemotherapy, hormone therapy, radiation therapy, and in some cases, targeted therapy drugs or immunotherapy. Breast cancer cells are hormone receptor-positive if they express either (or both) of the estrogen and/or progesterone receptors and are considered HER-2-positive if the breast cancer cells overexpress the protein, HER-2 (human epidermal growth factor receptor 2). If the cells meet none of these criteria, it is triple negative breast cancer (TNBC). TNBC makes up about 16% of all breast cancer diagnoses [93]. Few studies have examined the efficacy of MR in breast cancer models.

To investigate the effects of MR on breast cancer, a comprehensive study employed MR in a xenograft model for breast cancer, an immortalized human breast cell line, and an invasive breast cancer cell line [85]. In the animal model, athymic nude mice were injected with MCF10AT1 breast cancer cells. The control group was fed a diet containing 0.86% methionine while the MR group was fed a diet containing 0.12% methionine for 12 weeks [85]. Methionine restriction inhibited tumor progression in the mice by decreasing cell proliferation and increasing apoptosis [85]. MR increased expression of p21, but not p27, in the mouse mammary gland. Studies in the breast cancer cell line and immortalized breast cell line supported the involvement of p21 in the mechanism of action of MR. Additional mechanisms proposed involved the MR-induced reduction in circulating insulin and IGF1, which have both been linked to tumor growth, and the MR-induced depletion of polyamines [85]. 

TNBC has fewer treatment options than other forms of breast cancer due to the lack of a response to hormone therapy and drugs that target HER-2. Methionine restriction may provide a way to enhance efficacy of potential treatment options for TNBC. Tumor necrosis factor (TNF)-related apoptosis-inducing ligand (TRAIL) receptor agonists are an exciting possibility for cancer treatment due to their ability to induce apoptosis in cancer cells while having little effect on normal cells. Despite their efficacy in preclinical studies, TRAIL receptor agonists have not been successful in human clinical trials. Methionine deprivation enhances expression of TRAIL-R2 in TNBC cells, but not in normal breast epithelial cells [81]. Further, a methionine-free diet suppresses breast cancer growth and enhances the efficacy of the TRAIL-receptor 2 monoclonal antibody, lexatumumab, in inhibiting breast cancer growth in mice [81]. Studies have identified that out of 10 essential amino acids tested, depletion of methionine elicited the greatest inhibition of migration and invasion of TNBC cells [79]. Together, these data suggest that a combination of TRAIL receptor agonists with a methionine-restricted diet may enhance efficacy of this treatment [81]. 

Since some studies have shown that MR activates the integrated stress response [80] (a potentially detrimental action), recent studies have been undertaken to determine if blocking the ability of MR to activate the integrated stress response will enhance its efficacy in treating TNBC [94]. MR has been shown to activate two kinases involved in initiating the integrated stress response: general control nonderepressible 2 (GCN2) and protein kinase R-like endoplasmic reticulum kinase (PERK) [16]. It was postulated that blocking either or both of these kinases may enhance efficacy of methionine restriction in TNBC [94]. However, when either or both GCN2 and PERK were blocked, MR still initiated the integrated stress response in TNBC cells as evidenced by phosphorylation of eukaryotic initiation factor 2α (eIF2α), the downstream target of these kinases and induction of activating transcription factor 4 (ATF4), a downstream target of eIF2α [94]. Blocking the action of GCN2 and PERK did not affect the ability of MR to inhibit growth and induce apoptosis in TNBC cells [94]. Elucidating the mechanism by which MR activates the integrated stress response might provide novel targets to enhance efficacy of MR in slowing cancer growth [94].

### 4.4. Methionine Restriction and Colorectal Cancer

Colorectal cancer is the fourth leading cause of cancer-related death in the U.S. The relationship between methionine restriction and colon cancer has been established in tumor prevention and treatment in animal and cell culture models. A recent study evaluated the effects of MR on two patient-derived xenograft (PDX) models of colorectal cancer [12]. Methionine restriction (86% methionine-restricted diet) was initiated either two weeks before inoculation to determine the effects of MR on cancer prevention, or when the tumor became palpable to test the treatment effects of MR [12]. Initiating MR prior to tumor inoculation resulted in significantly impaired tumor growth in both PDX models. Initiating MR at the time the tumor was palpable significantly reduced tumor growth in one PDX model and tended to reduce tumor growth in the other, but this did not reach statistical significance [12]. Importantly, MR also enhanced efficacy of 5-fluorouracil, a chemotherapeutic drug with great efficacy against colorectal cancer [95], in the PDX model [12].

Inhibition of colon carcinogenesis was investigated using an azoxymethane-induced colon carcinogenesis rat model [86]. F344 rats were fed either control or 80% methionine-restricted diets for one week and were administered azoxymethane for two weeks to initiate colon carcinogenesis. Ten weeks after initiation of colon carcinogenesis (13 weeks after the start of the study), rats were sacrificed for examination of aberrant crypt foci (ACF) formation in the colon. Dietary MR, when maintained throughout the duration of the study, inhibited formation of large colonic ACF, which are well correlated with tumor formation, by over 80% [86]. Interestingly, consuming the MR diet for just the post-initiation period, had similar effects of consuming MR throughout the full 13-week study, with a 98% inhibition of large ACF formation in the colon. However, consuming the MR diet for just the initiation period (one week prior to- and two weeks during the azoxymethane treatment) had little effect on ACF formation, suggesting that the greatest inhibitory effects of MR on colon carcinogenesis occur primarily during the post-initiation phases [86]. MR-induced reduction in cell proliferation was hypothesized as a possible mechanism based on the 12% reduction in proliferation markers, BrdU and PCNA [86]. This study displayed MR’s inhibitory effect on colon carcinogenesis, presenting MR as a preventative treatment [86]. 

## 5. Translational Potential of Methionine Restriction as Treatment for People with Cancer

A growing body of literature conducted in cell culture and animal models indicates efficacy and safety of methionine restriction, alone or in combination with standard cancer therapies, as a treatment or preventative for a number of different cancers. However, there have been few studies investigating the anti-cancer potential of MR in humans. The provision of an MR diet to humans usually involves the use of an elemental amino acid medical beverage lacking methionine to ensure patients obtain desired quantities of protein since other food sources of protein must be limited to ensure adequate restriction of dietary methionine. A feasibility study published in 2002 established that methionine restriction, provided enterally, for an average of 17 weeks is safe and feasible in patients with advanced metastatic cancer (*n* = 8 subjects analyzed) [88]. A seven day study indicated that short-term MR, achieved through total parenteral nutrition, enhanced efficacy of 5-fluorouracil in patients with advanced gastric cancer [87]. Cystemustine is a drug that is most effective against human glioma and melanoma [96]. A phase II clinical trial of 22 patients (20 with metastatic melanoma and two with recurrent glioma) evaluated efficacy and safety of methionine restriction combined with conventional cystemustine treatment [89]. In this study, patients consumed a methionine-free diet for one day every two weeks during the two-month cystemustine treatment period [89]. The authors confirmed the feasibility of implementing a methionine-restricted diet in patients with advanced cancer and showed that following this MR feeding regimen did not negatively affect the patients’ nutritional status, as evidenced by stable body weights and relatively stable serum albumin and prealbumin concentrations [89]. However, the inclusion of MR once every two weeks had no clinically meaningful effects on survival [89]. 

The evidence produced from the limited clinical trials of MR in cancer patients, bolstered by the robust evidence of efficacy of MR in preclinical rodent and cell culture studies, potentiates MR as an exciting tool in cancer treatment. Additional large-scale clinical trials are needed to identify optimal treatment regimens of MR alone or in combination with standard care in different types and stages of cancer.

## 6. Conclusions

Dietary methionine restriction reduces circulating methionine concentrations in most human and animal studies within just days of consuming the diets. These data suggest that the MR-induced inhibition of cancer progression is due, at least in part, to cell-autonomous effects on tumors [12]. There are many alterations in methionine metabolism associated with cancer and understanding how dietary methionine affects cancer could yield important insights into novel treatment approaches [97]. Preclinical studies indicate that MR is effective when implemented in both prevention and treatment contexts and enhances efficacy of some standard cancer therapies. Studies have shown that MR exhibits anti-cancer activities in many types and stages of cancer. Studies identifying optimal degrees of methionine restriction are warranted due to the fact that MR is usually accompanied by weight loss, a symptom typically associated with poorer prognosis in cancer patients. To understand whether the preclinical successes of MR will translate to humans, larger scale clinical trials combining dietary methionine restriction with current standard treatment approaches in people with cancer are needed.

Progress has been made in identifying the mechanism by which MR inhibits cancer growth. Inhibition of thymidylate synthase activity, reduced polyamine biosynthesis, induction of apoptosis, and alterations in DNA methylation and glutathione formation are likely targets of MR, depending on the type of cancer. As the field moves toward implementation of MR in people with cancer, molecular studies into the mechanism of action of MR using cell culture and animal models will be important. For instance, identifying the mechanism by which MR inhibits thymidylate synthase activity and protein expression could yield a novel drug target. Given that normal cells can synthesize sufficient methionine for growth, but many cancer cells require exogenous methionine for survival, methionine restriction has potential as a cancer therapeutic [98]. Elucidating the signaling pathways and molecular mechanisms of methionine restriction could improve current therapeutic options and lead to the development of new targeted treatment options for people with cancer.

## Figures and Tables

**Figure 1 nutrients-12-00684-f001:**
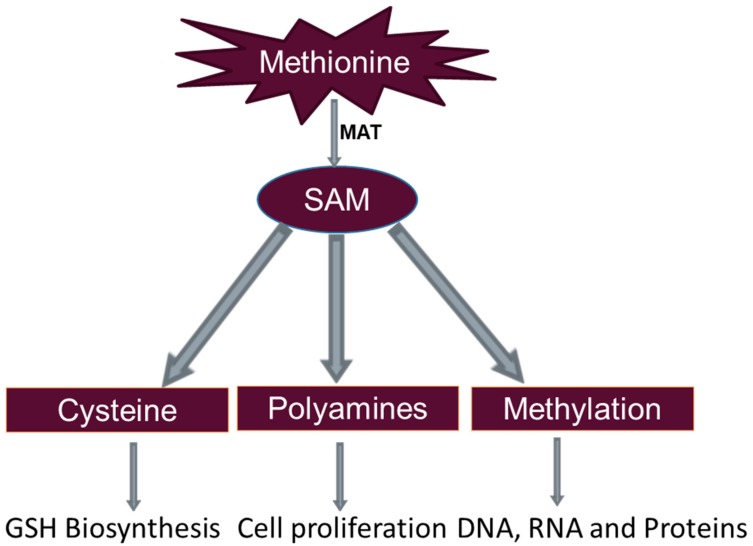
Methionine metabolism and major functions with relevance to GSH biosynthesis, Cell proliferation and methylation. GSH: Glutathione; MAT: Methionine adenosyltransferase; SAM: S-adenosyl methionine.

**Table 1 nutrients-12-00684-t001:** Methionine restriction and cancer biology.

Cancer Model	Effects of Methionine Restriction	Reference
**Cell culture**		
23 cancer cell lines: lung, bladder, prostate, cervical, colon, sarcoma, glioblastoma, melanoma, neuroblastoma, others	11 cell lines are absolutely dependent on methionine for growth.	[10]
21 human patient-derived tumors of various cancer types	Out of 21 human tumors, five (colon, breast, ovary, prostate, and a melanoma) were deemed methionine-dependent based on cell cycle analysis.	[11]
Tissue cultures of rat breast cancer, mouse lymphatic leukemia, human monocytic leukemia, rat liver epithelial cells, rat liver fibroblasts, mouse skin fibroblasts, human breast fibroblasts, human prostate fibroblasts	Normal cells can grow in methionine-depleted, homocysteine-supplemented media, while cancer cells cannot survive.	[71]
CNS tumor cell lines, fibroblast, and medulloblastoma cell lines	MR caused the following: (1) a marked increase of GADD45α and γ in the wt-p53 cell lines SWB61; (2) an increase in GADD34 and p21 protein in all of the methionine-dependent lines; and (3) the induction of MDA7 and phospho-p38 in DAOY and SWB39, consistent with marked transcriptional activation of the former under methionine stress.	[72]
Human prostate cancer cell line, primary prostate epithelial cells	MR synergistically enhances the anti-tumor effect of 5-FU by depletion of reduced folates, selective inhibition of thymidylate synthase (TS), and creation of an imbalanced nucleotide pool.	[74]
Human prostate cancer cell line	MR in DU145 and PC3 cells reduces mitochondrial membrane potential and induces caspase-dependent and -independent apoptosis.	[75]
Human prostate cancer cell line	MR inhibits phosphorylation but not protein expression of FAK and ERK in PC3 cells.	[76]
Human prostate cancer cell line	MR led to an accumulation of the cyclin-dependent kinase inhibitors p21 and p27.	[77]
Human prostate cancer cell line, human cervical carcinoma cell line	MR induces apoptosis of prostate cancer cells via the c-Jun N-terminal kinase-mediated signaling pathway.	[78]
Human TNBC cell line	Methionine deprivation increases the sensitivity to potential cancer drug in triple-negative breast cancer cells by enhancing TRAIL receptor-2 expression.	[79]
Human TNBC cell line and mouse model of TNBC	Methionine deprivation inhibited the migration and invasion of cancer cells. In addition, methionine deprivation reduced the activation of FAK and the expression of matrix MMP-2 and MMP-9.	[80]
Human TNBC cell line and mouse fibroblast	MR inhibited growth and induced apoptosis in TNBC cells in a GCN2- and PERK-independent mechanism.	[81]
**Animal models**		
Two patient-derived xenograft models of colorectal cancer and one mouse model of autochthonous soft-tissue sarcoma	Methionine restriction effectively inhibits tumor growth in two chemotherapy-resistant colorectal cancer samples from human patients. In addition, MR also suppresses tumor development in a mouse model of autochthonous soft-tissue sarcoma.	[12]
Sprague-Dawley rats with subcutaneously transplanted Walker tumor	Methionine restriction suppressed tumor growth.	[70]
Transgenic Adenocarcinoma of the Mouse Prostate (TRAMP)	Methionine restriction inhibits prostatic intraepithelial neoplasia in TRAMP mice.	[82]
Human gastric cancer xenograft in nude mice	Methionine depletion increased the 5-FU antitumor activity by modulating intratumoral folate metabolism.	[83]
Rats with Yoshida Sarcoma	Methionine deprivation inhibits tumor growth and metastasis with administration of 5-FU.	[84]
Mice injected with human pre-malignant breast epithelial cell line	Methionine restriction inhibits growth of breast tumors by increasing cell cycle inhibitors in nude mice.	[85]
F344 rats treated with azoxymethane to induce colon cancer	Methionine restriction inhibits colonic tumor development during post-initiation phases of carcinogenesis partially due to proliferation inhibition.	[86]
**Human Studies**		
Fourteen people with advanced gastric cancer	Methionine-depleting total parenteral nutrition had synergistic effects with 5-FU on human gastric cancer progression and TS activity.	[87]
Eight people with various metastatic solid tumors (renal cell carcinoma, carcinoid, sarcoma, pancreatic adenocarcinoma, prostate adenocarcinoma, follicular lymphoma)	Enterally-delivered MR reduced plasma methionine concentrations and is safe and tolerable in human patients with metastatic solid tumors.	[88]
Twenty-two people; 20 with metastatic melanoma and two with recurrent glioma	Methionine restriction was well tolerated (i.e., there was no indication of toxicity or nutritional concerns) but demonstrated little effect on survival.	[89]

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
