# Peer review of "Methionine Restriction and Cancer Biology"

_nutrients, 2020, doi:10.3390/nu12030684_

Round 1
Reviewer 1 Report
The authors have extensively discussed the role of methionine and restricting its amount as a potential therapy for cancer in this review article. The article is well written, easy to understand and the authors have been quite thorough in curating articles highlighting the advantage of methionine restriction in limiting cancer cell growth. There are few aspects, mentioned below, that could be improved:
- Sanderson et al., 2019. Methionine metabolism in health and cancer: a nexus of diet and precision medicine. I think this paper should be cited so readers can refer to it to understand methionine metabolism as a prequel to this manuscript which highlights methionine restriction and its effect in different cancers.
- In Figure 1, should it be SAM instead of SAMe?
- Line 179: The sentence, ‘The chemotherapy drug… ‘can be paraphrased to make it clearer.
- Line 199 and 219: Should it be ‘p21 and p27’ instead of ‘P21 and P27’?
- Rajanala SH, et al. 2019. Methionine restriction activates the integrated stress response in triple-negative breast cancer cells by a GCN2- and PERK-independent mechanism. It would be valuable to briefly discuss this paper as well in the breast cancer section.
- Line 265: Period after the reference.
- Either in the ‘Translational Potential’ or in the ‘Conclusions’ section it could be stressed that understanding how methionine restriction works would help improve targeted therapy.
- The cancer model and effects table in the end is extremely useful and would be good to mention this table in the main text, perhaps in the cancer overview section (Line 137).
Author Response
General Feedback:
The authors have extensively discussed the role of methionine and restricting its amount as a potential therapy for cancer in this review article. The article is well written, easy to understand and the authors have been quite thorough in curating articles highlighting the advantage of methionine restriction in limiting cancer cell growth. There are few aspects, mentioned below, that could be improved:
Specific Points:
- Sanderson et al., 2019. Methionine metabolism in health and cancer: a nexus of diet and precision medicine. I think this paper should be cited so readers can refer to it to understand methionine metabolism as a prequel to this manuscript which highlights methionine restriction and its effect in different cancers.
We agree that this thorough review on methionine metabolism and its relevance to cancer treatment is important to cite. We’ve included a reference to this paper in line 312.
- In Figure 1, should it be SAM instead of SAMe?
We have updated Figure 1 to indicate “SAM” instead of “SAMe”.
- Line 179: The sentence, ‘The chemotherapy drug… ‘can be paraphrased to make it clearer.
We expanded upon this sentence to help clarify the message. These edits can be found in lines 178-187.
- Line 199 and 219: Should it be ‘p21 and p27’ instead of ‘P21 and P27’?
We appreciate the reviewer catching these errors. We have updated the document to indicate “p21” and “p27” (now found in lines 202 and 222).
- Rajanala SH, et al. 2019. Methionine restriction activates the integrated stress response in triple-negative breast cancer cells by a GCN2- and PERK-independent mechanism. It would be valuable to briefly discuss this paper as well in the breast cancer section.
We thank the reviewer for bringing this to our attention. We’ve added a paragraph discussing this paper and feel its inclusion has strengthened our review article. The paragraph can be found in lines 240-252 in the breast cancer section.
- Line 265: Period after the reference.
We appreciate the reviewer catching this error. We have moved the period to after the reference.
- Either in the ‘Translational Potential’ or in the ‘Conclusions’ section it could be stressed that understanding how methionine restriction works would help improve targeted therapy.
We modified the Conclusions section to stress that understanding how methionine restriction works would improve current therapeutic options and potentially lead to development of new treatment options for people with cancer. The conclusions section has been extensively revised, with edits related to this comment being found in both paragraphs in the Conclusions section.
- The cancer model and effects table in the end is extremely useful and would be good to mention this table in the main text, perhaps in the cancer overview section (Line 137).
We thank the reviewer for his/her feedback. We have added a reference to the table at the end of the Methionine Restriction and Cancer Overview section (line 162).
Reviewer 2 Report
The review is very interesting and well written; however, the conclusions are too short. The conclusions can be enhanced by adding input on possible studies to perform. One minor issue than can also be addressed is how keto diets could affect mitochondrial metabolism and affect methionine related pathways.
Author Response
- The review is very interesting and well written; however, the conclusions are too short. The conclusions can be enhanced by adding input on possible studies to perform.
We thank the reviewer for his/her thoughts and agree that discussing future directions is an important part of the Conclusions section. We have expanded the Conclusions section to include our thoughts on important studies that should be performed. The conclusions section has been extensively revised, with major edits related to this comment being found in lines 314-319 and 323-326.
- One minor issue than can also be addressed is how keto diets could affect mitochondrial metabolism and affect methionine related pathways.
We appreciate the reviewer’s thoughts on this and agree that the ketogenic diet and mitochondrial metabolism are important considerations regarding methionine-related pathways and cancer. The current review is already broad and covers many topics including methionine metabolism, methionine restriction (MR), several types of cancers, and several mechanisms. If the reviewer agrees, in order to keep the review article focused on MR and cancer, we would rather not branch off into this topic.